# Knowledge of the risks, hygiene habits and perceived health issues associated with pesticide mixing and application among smallholder farmers

Shade J. Akinsete[1], Oluwaseun T. Adejumo[1], Mumuni Adejumo[1], Haruna Musa Moda[2], Stella Ibifunmilola[3]*

**1** Department of Environmental Health Sciences, Faculty of Public Health, University of Ibadan, Nigeria,
**2** Department of Environmental Health and Safety, University of Doha for Science and Technology, Qatar,
**3** Department of Health Professions, Manchester Metropolitan University, United Kingdom

* Stella.Ibifunmilola@mmu.ac.uk

## Abstract

Smallholder farmers' exposure to pesticide can be minimized by their hygiene behaviour during pesticide application. The aim of this study was to determine knowledge of the risks, hygiene habits and perceived health issues associated with pesticide mixing and application among smallholder farmers in Nigeria. A cross-sectional study was conducted on 162 smallholder farmers from Ibarapa North Local Government Area, using a validated structured questionnaire through a two-stage sampling method. Data were analyzed using descriptive statistics and chi-square test at p = 0.05. Farmers' mean age was 42.4 ± 12.3 years, 85.0% were male while 32.9% had tertiary education. Most (95.1%) farmers acknowledged that pesticides affect human health and 63.1% read, understood and followed pesticides label. Notably, only 4.9% acknowledged that banned or restricted pesticides cannot be used. Respondents identified pesticide exposure routes as inhalation (74.7%), dermal (15.4%), oral (1.9%) and eye contact (0.6%). Mean knowledge score was 7.9 ± 2.7 and 54.3% had poor knowledge of pesticide risk. Majority of farmers had direct skin contact with pesticide (83.3%) while 21.3% wore pesticide contaminated farm cloth home. Respondents' use of personal protective equipment (PPE) during pesticide mixing and application were: goggles (10.1%), coverall (29.0%), head cover (22.9%) and gloves (41.8%). Additionally, unsafe disposal of empty pesticide containers on farm was common among the farmers (42.6%). Mean hygiene habit score was 5.8 ± 1.9 and 85.2% had unsafe hygiene habit. Respondents' (90.9%) who had poor knowledge of pesticide risk was significantly (p = 0.025) higher among those who had unsafe hygiene habit during pesticide mixing and application. Reported symptoms by farmers included: dizziness > itchy eye > headache = skin irritation > nausea = coughing, during or after pesticide mixing and application. Farmers' knowledge about pesticide risk and safe hygiene habit was poor. Farmers' health may be at risk,

**Data availability statement:** All relevant data are within the manuscript and its Supporting Information files.

**Funding:** The author(s) received no specific funding for this work.

**Competing interests:** The authors have declared that no competing interests exist.

hence appropriate hygiene habit and use of PPE should be strictly adhered to during pesticide mixing and application.

---

## 1. Introduction

Pesticides have been commonly used to promote food security among farming communities as viable means of crop protection to improve yield and quality [1–4]. However, there are concerns regarding the increased and indiscriminate use of pesticides especially in low– to middle–income countries [2,4,5] of which some of the pesticide in use are classed as highly hazardous pesticides (HHPs) hence banned or their use are highly restricted. Excessive use, poor farmers' knowledge, improper handling and unsafe practices of pesticides among farmers in Africa is becoming increasingly disturbing [3,5,6]. Lack of knowledge and awareness among farmers in Africa have led to frequent use of banned pesticides that include HHPs based on WHO risk classification system [3,6,7]. This action among farmers is further heightened by the inadequate regulatory mechanisms and weak enforcement within the region [3]. Relatedly, farmers that are constantly exposed to pesticides may not have adequate knowledge of its occupational risk associated with lack of adherence to safety precaution when handling these chemicals. Hence the need for proactive measure aimed at regulating the use of pesticides and strengthen the monitoring of banned substance still in circulation.

Owing to the frequent use of banned and or restricted pesticides, especially organochlorines (OCPs) and organophosphates (OPPs) residues have been reported in air [8,9], water [10], soil [8], sediment [10], blood [11,12], urine [13,14] and other secondary matrices [11] in Africa. These studies are growing evidence regarding the ubiquity of pesticides in the environment and its associated health impact. Furthermore, storage of pesticides including OPPs (e.g., Chlorpyrifos) and OCPs (e.g., Endosulfan) have been reported in or around farmer's residence in many low– to middle–countries [2,4], including Africa [14,15]. Other reported unsafe practices among farmers include the absence or inadequate use of personal protective equipment (PPE), improper disposal, and reuse of empty pesticides containers [16–19].

In addition, due to poor personal safety adherence during pesticide application, farmers are frequently exposed to pesticide residues and when either inhaled, ingested or via skin contact, such present acute or chronic effect to the human body at some point [20,21] thereby impacting on farmers overall quality of life. Several pathways have been established whereby pesticide exposure occurs among farmers, including during mixing, loading and farm application [4,6,21]. As a result of close contact, farmers and farm workers are at higher risk of exposure to pesticides [13,21,22], consequently constituting a vulnerable group whose health may be seriously compromised [21]. Despite the provision of local and international guidelines and legislation regarding safe use of pesticide, exposure among farmers and farm works especially in developing countries has become an important occupational risk partly due to poor hygiene practices. Previous studies reported poor hygiene practices and behaviors among rural farmers in southwest Nigeria [19]; farm workers

in southwest Ethiopia [23]; Cocoa farmers in Ghana [24]; and horticultural farmers in Meru County, Kenya [25]. Several factors linked to farmers poor hygiene adherence during handling and application of pesticide include socio-demographic factors [26], high cost of equipment maintenance and incompatibility and or unavailability of personal protective equipment [27].

Nigeria has been ranked as the largest importer of pesticide in Africa, during the 2020 farming season, 147,446 tons of pesticides were imported into the country [28]. The informal nature of farming activity in the country, makes it difficult to ascertain the frequency of pesticide application among farmers. Despite this shortcoming, vast number of individuals are involved in farming activities in Nigeria. Although unsafe pesticide handling practices among smallholder farmers have been documented in many regions, evidence from farmers in rural communities in Nigeria remains limited. This study contributes to closing this gap by assessing farmers' knowledge of pesticide risks, hygiene habits during mixing and application, and perceived health issues. It is imperative to comprehend these in order to establish content-specific and locally relevant interventions that speak to farmers' life experiences—an issue that is sometimes overlooked in the international literature. Hence the study aimed to determine farmers' knowledge regards occupational risks, hygiene habits and perceived health issues associated with pesticide mixing and application among smallholder farmers of Ibarapa North Local Government Area, in Nigeria.

## 2. Methodology

### 2.1. Study area

The study was carried out in Ayete and Tapa towns in Ibarapa North Local Government Area (INLGA), Oyo State, Nigeria. Ayete town (7° 32′ 34.296″N and 3° 13′ 21.468″E) is the headquarters of INLGA which is 72 km south west of Ibadan and 60 km north east of Abeokuta. This region is best described as undulating land scattered with hills, ridges, inselbergs and rock outcrops with pockets of low-lying plains and valleys [29]. The climate is tropical with annual rainfall ranging between 1500–2000 mm and the relative humidity is > 80% (morning) and between 50–70% (afternoon). The mean annual temperature of the area is about 27°C [29]. The INLGA is an agricultural area with farming as the main occupation of the inhabitants. The area is known for the production of cash crops, e.g., cocoa, citrus and oil palm, as well as arable crops, e.g., cassava, yam, maize, melon and various vegetables [30]. This study was conducted in November 2019 which was characterized by late rains.

### 2.2. Sampling procedure

A cross-sectional study involving a total of 162 consented farmers randomly selected from within these agrarian communities were interviewed. Interviews were conducted with participants who had engaged in farming activities and who had mixed and/or applied pesticides within three months prior to the data collecting period. Farmers that practice organic farming were excluded from the study. Consenting farmers who had not participated in pesticide mixing and/or application within the three months preceding the data collecting period were as well excluded. At the end, 173 farmers were assessed for their eligibility and 162 farmers who met the inclusion criteria were interviewed making the response rate of 93.6%.

### 2.3. Data collection instrument

A validated, interviewer administered questionnaire was used to collect information about farmers' knowledge and safety practice regards pesticides handling and application [5,31]. The questionnaire was designed to collect information on socio-demographic characteristics, history of pesticide use, 15-point scale knowledge of pesticide risk (Scores of >7 was rated as Good), 13-point scale hygiene habit of pesticides (Scores of >6 was classified as Good), and reported illness associated with the use of pesticides. Knowledge regards occupational risks was considered based on participants

awareness level of the potential hazards and health risks related to the exposure to pesticides during the mixing and application process based on the following indicators: awareness about the effect of pesticide exposure on human health, environment, and ability to read and follow pesticides instruction/ direction on the labels, and awareness that any banned or restricted pesticides cannot be used. Identification of pesticide route of exposure (e.g., inhalation, dermal, oral and eye contact) were included. Each of the item on the indicators was assigned 1-point making a total of 15-point. Knowledge score was categorized as good (scores >7) and poor (scores ≤7).

Hygiene habits assessed include farmers practices and behaviors aimed at reducing exposure to pesticide residues and contaminants during or after mixing and application. To this, 13-items were considered in the questionnaire that include habits such as eat while mixing or spraying, drinking during mixing or spraying, smoking while mixing or spraying, taking bath after mixing or spraying, washed farm cloth separately from others, skin contact with pesticide, wear farm cloth contaminated with pesticide home and personal protective equipment worn during mixing and application of pesticide. Points were assigned to each of the items, hygiene habit score was computed and rated as unsafe hygiene habit (scores ≤6) and safe hygiene habit (scores >6), respectively.

Perceived Health Issues is the subjective assessment or belief about the potential health effects associated with pesticide exposure during mixing and application and measured using questionnaire. The indicators used were the health symptoms experienced by the farmers' during or after mixing and application of pesticides in the last 3 month prior to the survey. These symptoms were headache, dizziness, coughing, skin irritation, itchy eye, fatigue, stomachache and nausea. The questionnaire was translated into Yoruba language (and back translated to English), and pre-tested at Asejire farm settlement, Ibadan, Oyo State. Cronbach's Alpha was used to test for the consistency of the questionnaire and the instrument was improved afterwards for its effectiveness.

## 2.4. Data collection methods

The farmers had an informal association however members neither follow any set standards of practice nor receive any specialized training about pesticide application other than learning as apprentices on the job. To establish rapport with the farmers, formal introduction and protocols were observed to reassure the association members and seek their support and permission that guarantee the study success. At the initial meeting organized, the association leaders solicited members towards the study and explained the study rationale and its associated benefit to the farmers. Two research assistants were trained on how to administer the questionnaire and ensure they both have good understanding regards interviewing skills, how to review questionnaire to ensure completeness and related ethical consideration during the administration of the research instrument prior to the start of the data collection process. For each administered questionnaire, the study objectives were explained to the participants while anonymity and confidentiality were assured and signed consents was obtained from individual farmers prior to the start of the questionnaire administration. Efforts were made to avoid respondent influencing each other regards their response choices and research assistants ensured that all the questionnaires were duly completed by respondents by crosschecking immediately.

## 2.5. Data management and analysis

Data collected was sorted, checked for completeness and accuracy, compiled, entered and analyzed using Statistical package for Social Sciences (SPSS) version 20. Categorical variables were presented using percentages while mean and standard deviation were used to present continuous variables. Chi square test was used to analyze the association between farmers' sociodemographic characteristics, knowledge category and farmers' hygiene habits. Ordinary logistic regression analysis was carried out to measure the influence of respondents' educational status, smoking habit, access to training, knowledge and attitude on safe pesticide handling and application practices. The statistical significance tests were set at $p = 0.05$.

## 2.6. Ethical consideration

The study was conducted in accordance with all ethical procedures and the protocol was approved by the Joint Ethical Committee of University of Ibadan and University College Hospital, Ibadan, Nigeria (UI/EC//19/0434). Permission to conduct the study was obtained from the farmers' groups in the selected communities (Tapa and Ayete) within Ibarapa North Local Government Area and informed consent was obtained from the eligible farmers. The respondents were informed of their right to withdraw from the study at any time and confidentiality of the information was maintained by using assigned codes. Additional information regarding the ethical, cultural, and scientific considerations specific to inclusivity in global research is included (see Checklist in S1 File).

## 3. Results

### 3.1. Socio-demographic characteristics of farmers and history of pesticide use

Socio-demographic characteristics of farmers and history of pesticide use are presented on Table 1. Farmers' mean age was 42.4±12.2 years and 51.2% were older than 40 years. The respondents were mostly male (84.6%) and approximately 85% were married. Nearly half (48.1%) of the farmers had secondary school education while 31.5% had tertiary education, only 7.4% identified as not having formal education. All responding farmers attest to the use of pesticides as means of farm pest control and 52.5% affirmed to pesticides application on monthly basis. Relatedly, 63.0% of the farmers reported using pesticides for ≤14 years and among these respondents, 62.3% applied pesticides more than eight times during each calendar year. Knapsack manual sprayer for pesticide application was the common method (98.8%) of pesticide application among the respondents.

### 3.2. Farmers knowledge of pesticide risk

Based on participants response regards their knowledge of pesticide, 88.9% agreed that frequent exposure to pesticides has potential to impact human health and 78.4% affirmed its associated environment impact (Table 2). Despite the level of awareness regards its health and environmental effect among the participants, 68.5% still consider pesticides as indispensable for high crop yield. Among the participants, 65.4% responded that they regularly read and follow the pesticide safety instructions. However, only 27.8% of the farmers were aware of class of pesticides that are banned or restricted for use. There were high degree of awareness regards entry route of pesticide into the human body where 74.7% confirmed inhalation as the major entry route. Dermal (15.4%), oral (1.9%) and eye contact (0.6%) were scored low. Overall mean knowledge score was 7.9±2.7 and more than half of the farmers (54.3%) had poor knowledge of pesticide risk (Table 2).

### 3.3. Hygiene habits during pesticide mixing and/or application

Information regards hygiene habits during handling of pesticides presented in Table 3 showed 83.3% of the respondents reported that they had skin contact with pesticide at some point and 41.3% stated that children usually assist in mixing, loading and application of pesticide. When asked how individual clean used farm cloths after pesticide application, 21.3% said they travel back home in their farm cloth contaminated with pesticide residue. Other hygiene habit during mixing and/or application of pesticides mentioned were: eating (3.8%), drinking (14.6%), and lack of spraying away from wind direction (56.3%). Also, majority of the farmers reported bathing immediately after mixing and spraying (74.7%) and washing spraying cloth separately from others (86.7%). The proportion of personal protective equipment worn by the farmers during pesticide mixing and application was considered low among the participant where only 47.5% affirmed to using nose mask (47.5%), protective boots (41.8%) and gloves (41.8%), respectively. Overall mean hygiene habit score was 5.8±1.9, 85.2% of the respondents had unsafe hygiene habit during pesticide mixing and/or application (Table 3).

**Table 1. Socio-demographic characteristics and history of pesticide use of farmers.**

| Socio-demography | Frequency | Percentage | Mean± SD |
|---|---|---|---|
| **Age-group (years)** | | | |
| ≤ 20 | 9 | 5.6 | |
| 21–30 | 15 | 9.3 | |
| 31–40 | 55 | 34.0 | 42.4±12.3 |
| ≥ 41 | 83 | 51.2 | |
| **Sex** | | | |
| Male | 137 | 84.6 | |
| Female | 25 | 15.4 | |
| **Marital status** | | | |
| Married | 137 | 84.6 | |
| Others* | 25 | 15.4 | |
| **Education Level** | | | |
| No formal education | 12 | 7.4 | |
| Primary | 21 | 13.0 | |
| Secondary | 78 | 48.1 | |
| Tertiary | 51 | 31.5 | |
| **Farmland Size (Acres)** | | | |
| < 5 | 95 | 58.6 | |
| 6–10 | 47 | 29.0 | |
| > 10 | 20 | 12.3 | |
| **Monthly Earning (₦)** | | | |
| < 20,000 | 67 | 41.4 | |
| 20,000–40,000 | 67 | 41.4 | |
| > 40,000 | 28 | 17.3 | |
| **Interval of pesticide application** | | | |
| Weekly | 16 | 9.9 | |
| Monthly | 85 | 52.5 | |
| Every 3 months | 61 | 37.7 | |
| **Duration of using pesticide** | | | |
| ≤ 14 years | 102 | 63.0 | |
| >14–29 years | 48 | 29.6 | |
| > 29 years | 12 | 7.4 | |
| **Number of annual pesticide application.** | | | |
| ≥8 times | 101 | 62.3 | |
| <8 times | 61 | 37.7 | |
| **Application equipment.** | | | |
| Backpack sprayer | 160 | 98.8 | |
| Open tractor | 2 | 1.2 | |

Others*= Single, Divorced, Widowed, Separated.

### 3.4. Pesticide storage and disposal practices

Table 4 present pesticides storage and disposal practices among participating farmers. From the result, pesticides storage in locked chemical store was common among the group (54.9%), open shed designated for pesticides (22.8%), living

**Table 2. Knowledge of pesticide risk.**

| Knowledge statements | Frequency (%) | Mean±SD |
|---|---|---|
| Pesticides affect human health | 144 (88.9) | |
| Pesticides usually affect the environment | 127 (78.4) | |
| Pesticides are indispensable for high crop yield | 111 (68.5) | |
| Farmers must read, understand and follow pesticides labels | 106 (65.4) | |
| Some pesticides are banned or restricted for use | 45 (27.8) | |
| Pesticides that are banned or restricted cannot be used | 8 (4.9) | |
| **Route of exposure** | | |
| Inhalation | 121 (74.7) | |
| Dermal | 25 (15.4) | |
| Oral | 3 (1.9) | |
| Eye contact | 1 (0.6) | |
| **Knowledge Category** | | |
| Good | 74 (45.7) | 7.9±2.7 |
| Poor | 8854.3) | |

**Table 3. Hygiene habits during mixing and application of pesticides.**

| Variables | Frequency | (%) | Mean±SD |
|---|---|---|---|
| **Hygiene Habits during mixing and application** | | | |
| Eat while mixing or spraying | 6 | 3.8 | |
| Drink while mixing or spraying | 25 | 14.6 | |
| Smoke while mixing or spraying | 3 | 1.9 | |
| Spray with the direction of wind | 89 | 56.3 | |
| Bath immediately after mixing or spraying | 118 | 74.7 | |
| Wash spraying cloth separately from others | 137 | 86.7 | |
| Spill pesticides during mixing | 101 | 63.9 | |
| Spill pesticides during application on farm | 134 | 84.8 | |
| Skin contacts with pesticide | 130 | 83.3 | |
| Children assist in mixing, loading and application | 64 | 41.3 | |
| Children assisting adequately protected | 38 | 23.5 | |
| Wear farm cloth contaminated with pesticide home | 33 | 21.3 | |
| **PPE worn during mixing and application of pesticides** | | | |
| Coveralls | 45 | 29.0 | |
| Protective boots | 66 | 41.8 | |
| Glasses and goggles | 16 | 10.1 | |
| Gloves | 66 | 41.8 | |
| Nose mask | 75 | 47.5 | |
| Hat/head cover | 36 | 22.9 | |
| **Practice Category** | | | |
| Safe hygiene practice | 24 | 14.8 | 5.8±1.9 |
| Unsafe hygiene practice | 138 | 85.2 | |

area (14.8%), open field (6.2%), in a refrigerator (0.6%), and animal house (0.6%) were other forms of storage practices mentioned. Information regard management of unused and leftover (mixed, diluted) pesticides revealed 22.8% disposed

**Table 4. Pesticide storage and disposal practices.**

| Variable | N | % |
|---|---|---|
| **Place of storage of pesticides** | | |
| Open shed just for pesticide | 37 | 22.8 |
| In the open field | 10 | 6.2 |
| Locked chemical store | 89 | 54.9 |
| Living area | 24 | 14.8 |
| Refrigerator, with other items | 1 | 0.6 |
| Animal house | 1 | 0.6 |
| **Management of the unused leftover (Mixed, diluted) pesticides** | | |
| Dispose in the field | 37 | 22.8 |
| Mix only needed pesticides | 35 | 21.6 |
| Apply on other crops | 63 | 38.9 |
| Missing system | 28 | 16.2 |
| **Management the expired pesticide stocks** | | |
| Return to retailer | 22 | 13.6 |
| Hazardous waste collection sites | 5 | 3.1 |
| Dispose in the field | 16 | 9.9 |
| Buy only amount needed | 104 | 64.2 |
| Missing system | 15 | 9.3 |
| **Handling of the empty pesticide containers** | | |
| Discard on farm | 69 | 42.6 |
| Place in trash or dumpster | 18 | 11.1 |
| Burn on farm | 37 | 22.8 |
| Hazardous waste collection sites | 16 | 9.9 |
| Bury on-farm | 6 | 3.7 |
| Reuse for other purposes | 16 | 9.9 |

excess in the field or nearby surrounding, 38.9% said they reapply excess on already spread crops, while 21.6% said they only mix required amount to minimize excess. Concerning the handling of the expired pesticides stocks, 64.2% stated that they bought only the amount needed, 13.6% said the return to retailer while 9.9% said they dispose same in the field. Three major methods of handling empty pesticide container mentioned by farmers were to either discard on the farm (42.6%), burned on the farm (22.8%) and place in trash/dumpster (11.1%) respectively.

### 3.5. Reported illness experienced after Mixing and Application of Pesticides in the Last 3 months

Fig 1 showed the reported health symptoms experienced by the farmers during or after mixing and application of pesticides in the last 3 month. The reported symptoms followed the order: dizziness > itchy eye > headache = skin irritation > nausea = coughing. Other symptoms less reported were fatigue and stomach ache.

### 3.6. Comparison between Sociodemographic characteristics, knowledge category and farmers' hygiene habits

Safety practices score varied by farmers' sex, educational level, farm size, types of crops grown and knowledge category (Table 5). There was no significant association between safety practices and farmers' sex, educational level and farm size, respectively. However, proportion of farmers that grow vegetable was significantly higher among those who had unsafe hygiene habit during pesticide mixing and application (p = 0.045). Similarly, high proportion (90.9%) of farmers who had poor knowledge of pesticide risk was significantly higher among those who had unsafe hygiene

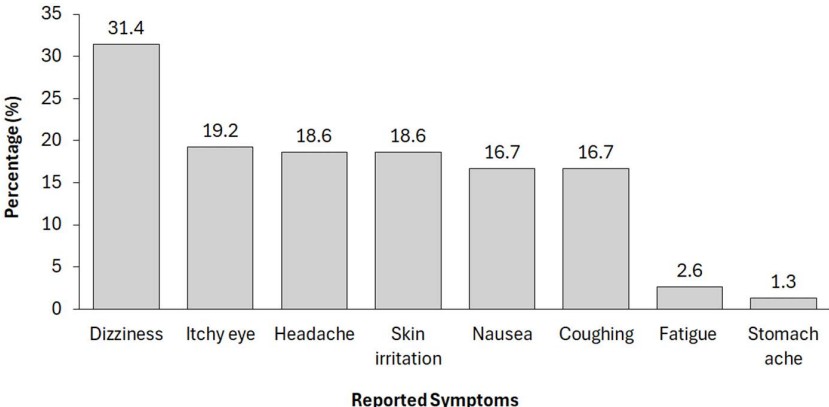

**Fig 1. Reported symptoms experienced after mixing and application of pesticides in the last 3 months.**

**Table 5. Comparison between Sociodemographic characteristics, knowledge category and farmers' hygiene habits.**

| Variables | Hygiene habits | | Total (%) | χ2 (Fisher's Exact) | p-value |
|---|---|---|---|---|---|
| | Safe (%) | Unsafe % | | | |
| **Sex** | | | | | |
| Male | 21 (15.3) | 116 (84.7) | 137 | 0.1860 | 0.667 |
| Female | 3 (12.0) | 22 (88.0) | 25 | | |
| **Educational level** | | | | | |
| No Formal education | 2 (16.7) | 10 (83.3) | 12 | 2.073 | 0.557 |
| Primary | 5 (23.8) | 16 (76.2) | 21 | | |
| Secondary | 9 (11.5) | 69 (88.5) | 78 | | |
| Tertiary | 8 (15.7) | 43 (84.3) | 51 | | |
| **Farm size (acres)** | | | | | |
| ≤ 5 | 17 (19.1) | 72 (80.9) | 89 | 3.549 | 0.170 |
| 6-10 | 3 (6.8) | 41 (93.2) | 44 | | |
| > 10 | 4 (14.8) | 25 (86.2) | 29 | | |
| **Type of crop grown** | | | | | |
| Cereals | 3 (27.3) | 8 (72.7) | 11 | 6.182 | 0.045 |
| Root crops | 6 (30.0) | 14 (70.0) | 20 | | |
| Vegetables | 15 (11.5) | 116 (88.5) | 131 | | |
| **Knowledge category** | | | | | |
| Poor | 8 (9.1) | 80 (90.9) | 88 | 5.001 | 0.025 |
| Good | 16 (21.6) | 58 (78.4) | 74 | | |

habit during pesticide mixing and application (p = 0.025). Ordinary logistic regression analysis of the respondents' types of crops grown, knowledge category and hygiene habits are presented in Table 6. The data showed that farmers who cultivate vegetables (OR = 2.195; C.I = 1.073–8.146) were more likely to practice safe hygiene habit during pesticide mixing and application than those who cultivated root crops and cereals. Also, participants who have good knowledge of pesticide risks presented better safe hygiene habit during pesticide mixing and application (OR = 2.759; C.I = 1.107–6.877).

**Table 6. Ordinary logistic regression analysis of the respondents' types of crop grown, knowledge category and farmers' hygiene habits.**

| Types of crop grown, knowledge category | ß | Sign. | Exp(ß) | Lower Bound | Upper Bound |
|---|---|---|---|---|---|
| **Type of crop grown** | | | | | |
| Cereals | R.C | R.C | 1.000 | R.C | R.C |
| Root crops | 0.134 | 0.873 | 1.143 | 0.223 | 5.866 |
| Vegetables | 0.929 | 0.038* | 2.195 | 1.073 | 8.146 |
| **Knowledge category** | | | | | |
| Poor | R.C | R.C | 1.000 | R.C | R.C |
| Good | 1.051 | 0.029* | 2.759 | 1.107 | 6.877 |

R.C = Reference Category; * Significant at 5%.

## 4. Discussion

This study determined risk, hygiene habits and perceived health issues associated with pesticide mixing and application among smallholder farmers in Nigeria. The study found that males dominated farming practice as with previous studies that reported similar findings [5,15,31–35]. This could be attributed to the task and intensive nature of farming activities hence less engaged in by females [5]. Most of the farmers in this study had formal education especially at the secondary and tertiary levels, compared with other studies that reported many farmers as illiterate or with little formal education [5,6,31]. Formal education and training are essential to enable farmers read, understand and perform some critical tasks such as calibration of sprayers and mixing of pesticides during pesticide application on the farm correctly. The high proportion of farmers that demonstrate awareness regards inhalation as a major route of exposure to pesticides is comparable to other studies in Tanzania [15] and Kuwait [31]. Fewer farmers compared to previous studies reported lack of knowledge of any route of exposure to pesticides [5,15].

Generally, the level of knowledge regards pesticide associated health risk among the sampled group in this study was generally considered as poor this further support earlier outcome reported in a previously [31]. In furtherance limited use of PPE during mixing and application of pesticides was affirmed by the participant which further corroborate other findings that reported on low use of PPE during handling and application of pesticides among farmers in other developing countries that include Tanzania [15], Ethiopia [5,6] and Bhutan [36]. Apart from cost of procuring recommended PPE, discomfort experienced during to prolong activity especially in hot and humid conditions has been identified as the main reason for farmers failure and reluctance to use PPE [5,31,37]. Poor use of PPE leads to inadequate protection during pesticide use with the implication for dermal absorption resulting from splashes and spills during mixing, loading and application pesticides [22]. Different types of PPE provide complementary levels of personal protection against dermal exposure, as such, use of multiple types of PPE for reducing exposure is required for highly toxic pesticides [22,31]. However, the protective ability of any PPE depends on proper and appropriate use. The aforementioned is imperative especially in developing countries where more toxic pesticides are used extensively [22,37].

Outcome from the study revealed that high proportion of the farmers had poor safety practices during pesticide application. This might cause environmental and human health problems. Poor pesticide handling practices has been reported to lead to harmful residues in harvested produce, soil and water contamination [38]. Our findings indicated direct exposure to pesticides by most farmers from spillage during mixing and application on farm. Furthermore, higher exposure levels will result from direct skin contact with pesticides as demonstrated in this study. Dermal exposure has been reported as one of the main routes of exposure for agricultural pesticides especially when they are readily absorbed through skin contact [22,39]. As a result, direct contact should be avoided and appropriate PPEs that reduces skin exposure should be used [39]. However, this study like previous ones demonstrate low use of coveralls during

pesticide mixing or application [39]. Additional measures to reduce exposure to pesticides is bathing immediately after spraying which has been reported to reduce dermal exposure as well as pesticide metabolites in the urine [40]. Different studies have reported lower proportion of farmers observing personal hygiene after pesticide application. A high proportion of the farmers in this study bath at some point after pesticide application similar to another study [39]. On the other hand, a recent study reported a high proportion of conventional farmers who did not take a bath immediately after spraying had significantly higher pesticide parent compound residues and metabolites in their urine [40]. Furthermore, Mergia et al. [5] reported nearly all farmers (94%) in their study did not bath after mixing or spraying. Similarly, only less than half of the farmers in other studies in Ethiopia took their bath after spraying pesticides [6,35]. Not choosing to bath after spraying may be attitudinal, lack of amenity on farm site or lack of knowledge that pesticide exposure can occur via indirect skin contact from contaminated clothes.

Another unsafe practice reported among the participants is the indiscriminate disposal of empty pesticide containers on the farm, such practice could serve as secondary route of exposure as well as environmental contamination via surface runoff or leaching [5,15]. Around 10% of the farmers indicated that they reuse empty pesticide containers for other purposes. Such practices may represent a route of non-occupational exposure [15]. Generally, a high proportion (83.3%) of the farmers who reported skin contact with pesticides during mixing and application reflects poor farmer hygiene practice, causing farmers to be at increased risk of dermal exposure.

Farmers who had poor knowledge of pesticide risk was significantly higher among those who had unsafe hygiene habit during pesticide mixing and application. Conversely, farmers who have good knowledge of pesticide risks had better safe hygiene habit during pesticide mixing and application. This is an important empirical confirmation of a critical gap that could be addressed through targeted and context-specific education and training programs. Furthermore, pesticide-related health symptoms reported by respondents included dizziness, itchy eye, headache, skin irritation, nausea, coughing and others. Similar symptoms have been reported in previous studies in Kuwait [31], Ethiopia [5] and Nigeria [19]. However, a key limitation of the study is the reliance on self-reported information, which was not validated through observation or independent data sources. For example, reported use of protective clothing and self-described health issues were not verified by physical inspection or cross-checked against health center records. Consequently, the results may be affected by recall bias or social desirability bias. Future research should combine interview data with observational checks and health records to enhance accuracy.

## 5. Conclusion

Pesticides can have negative effects on the environment and human health if handled improperly, hence farmers' knowledge of the risks, hygiene habits and perceived health issues associated with pesticide mixing and application is crucial. Although, a high proportion of participants had formal education, the study revealed poor knowledge of pesticide risks and insufficient use of personal protective equipment during mixing and application of pesticides among farmers. The implication is that education without appropriate/content-specific training on pesticide safety may not translate to proper safe practices. Moreover, our findings indicated direct exposure to pesticides occurs from spillage during mixing and application on farm. Farmers in this study displayed poor hygiene practices by improper disposal of empty pesticides containers in the field as well as high prevalence of skin contact with pesticides during mixing and application. Thus, suggesting a potentially serious public health problem in this study area. Farmers who had poor knowledge of pesticide risk did not exhibit good safety practices during pesticide application. Farmers' may be at risk of having health challenges in the nearest future, hence the use of personal protective equipment should be strictly adhered to during pesticide handling.

## Supporting information

**S1 File. Checklist Inclusivity in global research questionnaire.**
(DOCX)

 

**S2 File. Database.**
(XLSX)

## Acknowledgments

The authors acknowledge farmers who gave informed consent and participate in the study.

## Author contributions

**Conceptualization:** Shade J. Akinsete, Haruna Musa Moda.

**Data curation:** Oluwaseun T. Adejumo, Mumuni Adejumo.

**Investigation:** Oluwaseun T. Adejumo.

**Methodology:** Shade J. Akinsete, Stella Ibifunmilola.

**Project administration:** Shade J. Akinsete, Haruna Musa Moda.

**Resources:** Oluwaseun T. Adejumo, Stella Ibifunmilola.

**Validation:** Shade J. Akinsete, Mumuni Adejumo, Haruna Musa Moda.

**Visualization:** Mumuni Adejumo, Stella Ibifunmilola.

**Writing – original draft:** Oluwaseun T. Adejumo, Mumuni Adejumo.

**Writing – review & editing:** Shade J. Akinsete, Haruna Musa Moda.

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
