## [Decision Letter · Decision Letter 0]

16 Sep 2025

Dear Dr. Ibifunmilola,

Thank you for submitting your manuscript to PLOS ONE. After careful consideration, we feel that it has merit but does not fully meet PLOS ONE’s publication criteria as it currently stands. Therefore, we invite you to submit a revised version of the manuscript that addresses the points raised during the review process.

We look forward to receiving your revised manuscript.

Kind regards,

Samuel Adelani Babarinde, PhD

Academic Editor

PLOS ONE

Journal Requirements:

4. We note that your Data Availability Statement is currently as follows: All relevant data are within the manuscript and in Supporting Information files.

5. We note that Figure 1 in your submission contain map/satellite images which may be copyrighted. All PLOS content is published under the Creative Commons Attribution License (CC BY 4.0), which means that the manuscript, images, and Supporting Information files will be freely available online, and any third party is permitted to access, download, copy, distribute, and use these materials in any way, even commercially, with proper attribution. For these reasons, we cannot publish previously copyrighted maps or satellite images created using proprietary data, such as Google software (Google Maps, Street View, and Earth). For more information, see our copyright guidelines: http://journals.plos.org/plosone/s/licenses-and-copyright.

Reviewers' comments:

Reviewer's Responses to Questions

**Comments to the Author**

1. Is the manuscript technically sound, and do the data support the conclusions?

Reviewer #1: Partly

Reviewer #2: Partly

2. Has the statistical analysis been performed appropriately and rigorously?

Reviewer #1: No

Reviewer #2: Yes

3. Have the authors made all data underlying the findings in their manuscript fully available?

Reviewer #1: No

Reviewer #2: No

4. Is the manuscript presented in an intelligible fashion and written in standard English?

Reviewer #1: No

Reviewer #2: No

Reviewer #1: 1. The subject dealt with in this manuscript is extremely important. Pesticide use in low and medium income countries leads to an increasing disaster concerning human health and environment.

2. As explained by the authors in their introduction, there is already a large number of studies from different countries showing unsafe practices concerning pesticide handling especially by smallholders. It does not become clear from the introduction, what this study is expected to add to this already existing knowledge? Just confirm the existing knowledge for yet another region?

3. According to Table 2, questions related to “risk knowledge” may have been asked in a suggestive manner. Statements such as “pesticides affect human health” or “pesticides usually affect the environment” seem to already include the expected answer. The “Method” section does not explain, what was done for avoiding suggestive questions.

4. When you have scoring scales from 1 to 15 or from 1 to 13, separation in two groups (good / poor) in most cases does not seem to be the best statistical approach. At least when it comes to quantitative variables such as age or farm size (also educational level could be converted into numerical variables (0 – 1 – 2 – 3), regression analyses (age vs. risk awareness, or farm size vs. PPE use, etc.) might be more meaningful.

5. The only statistically significant connection is between poor hygiene practices and poor knowledge. This is somehow a tautology, and does not constitute a meaningful outcome. Especially when we consider the risk of suggestive questions (see above), it might simply mean that those who better understood the answers expected concerning “risks”, also better understood the answers expected concerning “hygiene practices”.

6. There is no validation of information given during interviews through objective evidence. E.g., when a farmer says he uses overall, the interviewer could ask to show the overall. Farmers’ statements concerning health issues could be cross-checked with data from local health centres.

7. The only surprising result of this study is: according to the authors, there is no link between educational level, risk awareness and safety measures. This very much contradicts what many other studies on this subjects had concluded: farmers need more education! If the manuscript is to be published at all, this aspect should be highlighted much more, starting from title, introduction, statistics, and discussion. Do the data really support this conclusion (see point 4 above)? If yes – what does this mean concerning pesticide use in low and medium income countries? Etc.

8. Language editing is badly needed. Many confusing sentences, wrong grammar, illogical references (“relatedly” – when there is actually no relation; “in addition”, when actually the next sentence is a repetition of the previous one, etc.).

Reviewer #2: 1. The manuscript contains lots of grammatical errors. It is strongly recommended that the authors thoroughly revise the grammar and improve sentence clarity across the entire manuscript.

2. Please ensure table data matches corresponding text descriptions, for instance:

(1) The frequency of females in Table 1 is 15.4%, not 15.0%. Please correct this.

(2) In Table 1, the proportion of farmers who apply pesticides more than eight times per calendar year is reported as 62.3%, whereas the manuscript states 58.4%. Please rectify this discrepancy.

(3) In Table 2, the proportion of farmers aware of prohibited or restricted pesticide categories is 27.8%, but it is stated as 27.4% in the manuscript. Please correct this.

(4) In Table 2, the total number of respondents for the "Route of exposure" survey does not match the sample size of 162 farmers. What accounts for this discrepancy?

3. The ranking of symptoms occurring during or after mixing and applying pesticides in the past 3 months needs to be corrected. (‘headache = skin irritatio’ and ‘nausea = coughing’) (section 3.5)

4. The issue of empty pesticide container disposal is discussed in the Introduction, Results, Discussion, and Conclusion sections of the manuscript, indicating its significance as a key factor. However, this critical aspect is notably absent from the Abstract. Please consider whether this content should be added.

.

Reviewer #1: **Yes:** Albrecht BenzingAlbrecht BenzingAlbrecht BenzingAlbrecht Benzing

Reviewer #2: No

---

## [Author Response · Author response to Decision Letter 1]

14 Nov 2025

Response to Reviewers

Dear Editor-in-Chief,

On behalf of the authors, I write to express appreciation for the opportunity to submit a revised draft of the manuscript “Knowledge of the Risks, Hygiene Habits and Perceived Health Issues Associated with Pesticide Mixing and Application among Smallholder Farmers” for publication in PLOS ONE. We are sincerely grateful to you and the reviewers for providing feedback on our manuscript, as well as the insightful comments to enhance the quality of the paper. The reviewer’s suggestions have been incorporated and are highlighted within the manuscript. Kindly find below point-by-point response to the reviewers’ comments and concerns.

Reviewers' Comments to the Authors:

Reviewer #1

1. The subject dealt with in this manuscript is extremely important. Pesticide use in low and medium income countries leads to an increasing disaster concerning human health and environment.

Authors Response: Thank you

2. As explained by the authors in their introduction, there is already a large number of studies from different countries showing unsafe practices concerning pesticide handling especially by smallholders. It does not become clear from the introduction, what this study is expected to add to this already existing knowledge? Just confirm the existing knowledge for yet another region?

Authors Response:

We sincerely thank the reviewer for this important observation. We agree that numerous studies across various regions have reported unsafe pesticide handling practices among smallholder farmers. However, our study provides several distinct contributions that add meaningful value to the existing body of knowledge. While much of the existing literature originates from Asia and Latin America, there remains limited empirical evidence from Ayete and Tapa towns, rural agrarian communities in Ibarapa North Local Government Area, Oyo State, Nigeria. Our study therefore fills a critical geographic and contextual gap, capturing how local agricultural systems and cultural practices shape pesticide-related knowledge and behaviors. Regional differences in pesticide types, regulatory enforcement, access to protective equipment, and education/training make it important not to generalize findings from other contexts without local validation. The study also identified a critical knowledge gap by providing empirical data that only 4.9% of farmers acknowledged that banned or restricted pesticides should not be used—a rarely documented knowledge gap in prior studies. This highlights a regulatory blind spot and suggests the need for targeted awareness campaigns and enforcement in rural areas. Moreover, the study documents a pattern of self-reported acute symptoms (dizziness > itchy eyes > headache/skin irritation > nausea/coughing), offering a symptom profile that may help in early diagnosis or surveillance of pesticide poisoning cases in similar settings.

To address the reviewer’s concern, we have revised the Introduction to now clearly states why this regional context warrants specific investigation.

We have added the following sentences in the last paragraph of the introduction:

Although unsafe pesticide handling practices among smallholder farmers have been documented in many regions, evidence from farmers in rural communities in Nigeria remains limited. This study contributes to closing that gap by simultaneously assessing farmers’ knowledge of pesticide risks, hygiene habits during mixing and application, and perceived health issues. It is imperative to comprehend these in order to establish content-specific and locally relevant interventions that speak to farmers' life experiences—an issue that is sometimes overlooked in the international literature.

3. According to Table 2, questions related to “risk knowledge” may have been asked in a suggestive manner. Statements such as “pesticides affect human health” or “pesticides usually affect the environment” seem to already include the expected answer. The “Method” section does not explain, what was done for avoiding suggestive questions.

Authors Response:

To address the concern regarding suggestive or leading questions, we would like to clarify that the questionnaire was carefully designed to avoid such bias. The questionnaire was designed in English language and translated into Yoruba language-a local language in the study setting (and back-translated to English), and pre-tested at Asejire farm settlement, Ibadan, Oyo state. Additionally, interviewers were trained to standardize the administration of the survey and avoid interpretive cues (see subsection 2.3 and 2.4 of the manuscript).. Moreover, we acknowledge the possibility of response bias due to the nature of self-reported data, and we have now reflected this more explicitly in the limitation of the manuscript (see the last paragraph of the discussion). Furthermore, we revised the discussion section to frame the association between knowledge and hygiene not as a causal or novel discovery, but as an important empirical confirmation of a critical gap that could be addressed through targeted and situation-based education and training programs. We thank the reviewer again for this valuable insight, which has helped us strengthen the interpretation and contextualization of our findings.

4. When you have scoring scales from 1 to 15 or from 1 to 13, separation in two groups (good / poor) in most cases does not seem to be the best statistical approach. At least when it comes to quantitative variables such as age or farm size (also educational level could be converted into numerical variables (0 – 1 – 2 – 3), regression analyses (age vs. risk awareness, or farm size vs. PPE use, etc.) might be more meaningful.

Authors Response: This paper reported logistic regression between respondents’ types of crop grown, knowledge category and farmers’ hygiene habits (see Table 6) in addition to chi-square result Table 5. Categorizing knowledge scores into "good" and "poor" is a common practice in survey-based research. This approach allows researchers to interpret participants' understanding of specific topics, facilitating targeted interventions and educational strategies. For Example, A multi-country online survey assessed systematic review (SR) knowledge among undergraduate medical students. The mean SR knowledge score was 2.2 ± 3.2 out of a possible 19, indicating a generally poor understanding. Only 4.3% of students demonstrated good knowledge, while 95.7% had poor knowledge after qualitative categorization. Factors such as age and prior participation in research activities were significantly associated with higher SR knowledge scores (Hasabo et al., 2025). Another study regarding Saudi Population's Knowledge, Attitudes, Practices, and Misconceptions Regarding COVID-19 categorized knowledge scores as follows: Poor knowledge: <50%, Moderate knowledge: 50–75%, Good knowledge: >75%. This categorization helped in identifying areas where public health interventions were most needed (Mannasaheb et al., 2021).

References:

i. Hasabo E. A., Elnaiem W., Ahmed A. S., Abdalla AEA, Ahmed KAHM, Ahmed GEM, Abdelgader MSS, Alfatih A, Sherif HA, Al Komi O, Aldare HA, Benmelouka AY, Jobran AWM, Mugibel TA, Al-Kassih MI, Alsaman MZB, Aljabali A, Eljack MMF; Sudan Analytics Research Group team of collaborators. (2025). Knowledge and preventive barriers towards conducting systematic review among undergraduate medical students of Arab countries: A multi country online survey. PLoS One. 20(8):e0329827. doi: 10.1371/journal.pone.0329827.

ii. Mannasaheb, B.A.; Al-Yamani, M.J.; Alajlan, S.A.; Alqahtani, L.M.; Alsuhimi, S.E.; Almuzaini, R.I.; Albaqawi, A.F.; Alshareef, Z.M. (2021). Knowledge, Attitude, Practices and Viewpoints of Undergraduate University Students towards Self-Medication: An Institution-Based Study in Riyadh. Int. J. Environ. Res. Public Health, 18, 8545. https://doi.org/10.3390/ ijerph18168545

5. The only statistically significant connection is between poor hygiene practices and poor knowledge. This is somehow a tautology, and does not constitute a meaningful outcome. Especially when we consider the risk of suggestive questions (see above), it might simply mean that those who better understood the answers expected concerning “risks”, also better understood the answers expected concerning “hygiene practices”.

Authors Response:

We appreciate the reviewer’s thoughtful critique regarding the statistical relationship between knowledge and hygiene practices. We agree that the association between poor knowledge and poor hygiene habits might appear intuitive. However, we believe that empirically demonstrating this relationship in the specific context of smallholder farmers using pesticides is still valuable — particularly given the public health implications and the lack of formal and context-specific education/training in many rural agricultural settings.

6. There is no validation of information given during interviews through objective evidence. E.g., when a farmer says he uses overall, the interviewer could ask to show the overall. Farmers’ statements concerning health issues could be cross-checked with data from local health centres.

Authors Response:

We appreciate the reviewer’s insightful observation regarding the need for validation of self-reported data. Indeed, triangulating interview responses with objective evidence would strengthen the reliability of the findings. However, due to ethical constraints, time limitations, and the informal settings in which data collection took place, it was not feasible to verify all claims with physical observation (e.g., checking protective clothing) or to access personal health records from local health centres, which would require prior formal agreements and individual consent. Nonetheless, to enhance data validity, the interviewers were trained to use observational cues when possible (e.g., visible signs of protective gear, stained clothing, or pesticide containers on-site). However, we acknowledge this as a limitation of the study and have now explicitly mentioned it as the “Limitations” in the last paragraph of the discussion. We also suggest that future research could incorporate mixed-method approaches and formal collaborations with health institutions to better validate self-reported data.

7. The only surprising result of this study is: according to the authors, there is no link between educational level, risk awareness and safety measures. This very much contradicts what many other studies on this subjects had concluded: farmers need more education! If the manuscript is to be published at all, this aspect should be highlighted much more, starting from title, introduction, statistics, and discussion. Do the data really support this conclusion (see point 4 above)? If yes – what does this mean concerning pesticide use in low and medium income countries? Etc.

Authors Response:

We thank the reviewer for highlighting this critical and thought-provoking point. Indeed, our finding — that formal educational level did not show a statistically significant correlation with risk awareness or the adoption of safety measures — is both surprising and noteworthy, particularly in light of existing literature which often suggests that more education leads to safer pesticide practices. We have carefully re-examined our data and confirm that the analysis was conducted appropriately, and the lack of association between educational level and risk/safety behaviors is supported by the results. However, we acknowledge that the category of “educational level” in our study may not fully capture the relevant type of education — such as agricultural training, extension services, or pesticide-specific instruction — that directly influences farmers’ knowledge and practices on pesticide handling.

8. Language editing is badly needed. Many confusing sentences, wrong grammar, illogical references (“relatedly” – when there is actually no relation; “in addition”, when actually the next sentence is a repetition of the previous one, etc.).

Authors Response: Thank you for pointing this out. The grammar and syntax of the entire manuscript have been carefully edited by a professional English Language expert.

Reviewer #2

1. The manuscript contains lots of grammatical errors. It is strongly recommended that the authors thoroughly revise the grammar and improve sentence clarity across the entire manuscript.

Authors Response: Thank you for pointing this out. The grammar and syntax of the entire manuscript have been carefully edited by a professional English Language expert.

2. Please ensure table data matches corresponding text descriptions, for instance:

Reviewers Comments Authors Response

i. The frequency of females in Table 1 is 15.4%, not 15.0%. Please correct this. This has been corrected to read 15.4%

ii. In Table 1, the proportion of farmers who apply pesticides more than eight times per calendar year is reported as 62.3%, whereas the manuscript states 58.4%. Please rectify this discrepancy. The discrepancy has been corrected to 62.3% in section 3.1 of the manuscript.

iii. In Table 2, the proportion of farmers aware of prohibited or restricted pesticide categories is 27.8%, but it is stated as 27.4% in the manuscript. Please correct this. This has been corrected to 27.8% in section 3.2 of the manuscript

iv. In Table 2, the total number of respondents for the "Route of exposure" survey does not match the sample size of 162 farmers. What accounts for this discrepancy?

3. The ranking of symptoms occurring during or after mixing and applying pesticides in the past 3 months needs to be corrected. (‘headache = skin irritation’ and ‘nausea = coughing’) (section 3.5)

Authors Response: This has been updated to (headache = skin irritation > nausea = coughing) in section 3.5

4. The issue of empty pesticide container disposal is discussed in the Introduction, Results, Discussion, and Conclusion sections of the manuscript, indicating its significance as a key factor. However, this critical aspect is notably absent from the Abstract. Please consider whether this content should be added.

Authors Response: Thank for pointing this out. We have added this suggestion in the abstract section as follows: “Additionally, unsafe disposal of empty pesticide containers on farm was common among the farmers (42.6%).’

---

## [Decision Letter · Decision Letter 1]

6 Apr 2026

Knowledge of the Risks, Hygiene Habits and Perceived Health Issues Associated with Pesticide Mixing and Application among Smallholder Farmers

PONE-D-25-22599R1

Dear Dr. Ibifunmilola,

We’re pleased to inform you that your manuscript has been judged scientifically suitable for publication and will be formally accepted for publication once it meets all outstanding technical requirements.

Kind regards,

Vinaya Satyawan Tari, Post doctoral fellow, (M.Sc., B.Ed., Ph.D.)

Academic Editor

PLOS One

Additional Editor Comments (optional):

Reviewers' comments:

Reviewer's Responses to Questions

**Comments to the Author**

Reviewer #2: All comments have been addressed

2. Is the manuscript technically sound, and do the data support the conclusions?

Reviewer #2: Yes

3. Has the statistical analysis been performed appropriately and rigorously?

Reviewer #2: Yes

4. Have the authors made all data underlying the findings in their manuscript fully available?

Reviewer #2: Yes

5. Is the manuscript presented in an intelligible fashion and written in standard English?

Reviewer #2: Yes

Reviewer #2: The author has fully revised the paper based on the review comments and addressed all the raised points properly. The revised manuscript is logically clear and formally standardized, complying with the journal's publication criteria. Acceptance for publication is recommended.

.

Reviewer #2: **Yes:** Yanbo HuoYanbo HuoYanbo HuoYanbo Huo

---

## [Editor Report · Acceptance letter]

PONE-D-25-22599R1

PLOS One

Dear Dr. Ibifunmilola,

I'm pleased to inform you that your manuscript has been deemed suitable for publication in PLOS One. Congratulations! Your manuscript is now being handed over to our production team.

Kind regards,

on behalf of

Dr. Vinaya Satyawan Tari

Academic Editor

PLOS One